# A dynamic ensemble model for short-term forecasting in pandemic situations

**Jonas Botz**[1]*, **Diego Valderrama**[1], **Jannis Guski**[1], **Holger Fröhlich**[1,2]*

**1** Department of Bioinformatics, Fraunhofer Institute for Algorithms and Scientific Computing (SCAI), Sankt Augustin, Germany, **2** Bonn-Aachen International Center for IT, University of Bonn, Bonn, Germany

* jonas.botz@scai.fraunhofer.de (JB); holger.froehlich@scai.fraunhofer.de (HF)

**Data Availability Statement:** The data is publicly available under the following three links: https://github.com/robert-koch-institut https://github.com/googleapis/google-api-python-client https://www.

## Abstract

During the COVID-19 pandemic, many hospitals reached their capacity limits and could no longer guarantee treatment of all patients. At the same time, governments endeavored to take sensible measures to stop the spread of the virus while at the same time trying to keep the economy afloat. Many models extrapolating confirmed cases and hospitalization rate over short periods of time have been proposed, including several ones coming from the field of machine learning. However, the highly dynamic nature of the pandemic with rapidly introduced interventions and new circulating variants imposed non-trivial challenges for the generalizability of such models. In the context of this paper, we propose the use of ensemble models, which are allowed to change in their composition or weighting of base models over time and could thus better adapt to highly dynamic pandemic or epidemic situations. In that regard, we also explored the use of secondary metadata—Google searches—to inform the ensemble model. We tested our approach using surveillance data from COVID-19, Influenza, and hospital syndromic surveillance of severe acute respiratory infections (SARI). In general, we found ensembles to be more robust than the individual models. Altogether we see our work as a contribution to enhance the preparedness for future pandemic situations.

## 1. Introduction

In late 2019 a novel coronavirus SARS-CoV-2 emerged [1]. This not only gave rise to the COVID-19 pandemic but also affected every aspect of human life, from an economic downturn, and disruption in education and social interactions to severe health implications including millions of deaths [2–4]. Early on, governments struggled to find a balance between containing the spread of the virus and maintaining as much economy, social interactions, and educational services as possible. Important indicators for decision-making were in many countries the number of confirmed cases and the hospitalization rate. During that time many models were developed for short-term forecasting of the number of incident cases and hospitalizations, respectively [5], modeling strategies in this field include mechanistic, machine learning, and hybrid modeling strategies [5]. These models have been successful in capturing periods of the pandemic and for scenario planning. However, they often depend on fundamental parameters derived from historical data or expert knowledge. The highly dynamic

data.gouv.fr/fr/organizations/sante-publique-france/#/presentation.

**Funding:** This work has been supported by the AIOLOS (Artificial Intelligence Tools for Outbreak Detection and Response) project. The project was supported by the French State and the German Federal Ministry for Economic Affairs and Climate Action (grant number 01MJ22005A) and the French Ministry of Economy and Finance in the context of the France 2030 initiative and the Franco-German call on Artificial Intelligence technologies for risk prevention, crisis management, and resilience. Authors JB, JG and HF received funding from the German Federal Ministry for Economic Affairs and Climate Action. The funders had no role in study design, data collection and analysis, decision to publish, or preparation of the manuscript.

**Competing interests:** The authors have declared that no competing interests exist.

nature of the pandemic, characterized by the rapid introduction of non-pharmaceutical interventions, new vaccines, and emerging virus variants, presented significant challenges for the generalizability of all forecasting models over extended periods. Consequently, models had to be frequently re-calibrated to stay aligned with the current situation. Since each modeling technique unavoidably comes along with its own assumptions and limitations, ensemble models have been proposed for forecasting the spread of infectious diseases like Influenza [6–8] or Ebola [9] and later for COVID-19 [10, 11]. In principle, ensemble models can be understood as a collection of base models, which all produce an output based on each model's assumption plus an algorithm or meta-model that combines them into one ensemble output. The advantage of such an ensemble approach is that the bias of the individual models is reduced, making the final output more robust [12]. In the literature, such ensemble methods often use the mean (e.g., [11]) or median (e.g., [10]) of the base model outputs. However, pandemics like COVID-19 are dynamic, there are times when the number of cases barely changes, there is exponential growth and decay, and there are turning points of waves, which can all depend on external factors like interventions [13, 14], people's behavior [15], seasonality [16, 17], or variants of concern [18, 19].

To enhance the robustness of models against the influence of such exogenous factors and to be better prepared for future pandemics we here propose an ensemble modeling approach that is dynamically adjusted, to either select a specific model out of an ensemble at a given time or to weigh models' predictions according to the current situation by using a meta-model. To test our, approach we here implemented several auto-regressive models as base models: a log-linear regression, ARIMA [20], XGBoost [21], Random Forest [22], and an LSTM [23] model. We then first evaluated the performance of each base model and compared it to baseline ensemble methods using mean or median averaging. In the next step, we implemented a multi-layer perceptron (MLP) with softmax heads as a meta-model. The base models' forecasts and performances were used as input for the meta-model which was trained in either one of two ways: 1. select one of the models (selection), 2. combine the model's predictions into one prediction (stacking). In addition, we tested whether the inclusion of metadata coming from Google Trends could inform the meta-model to make better decisions.

## 2. Materials and methods

### 2.1 Surveillance data

We used different datasets related to COVID-19: the daily number of confirmed incident cases, hospital admissions (hospitalization), and deaths for Germany and France, respectively. Additionally, we evaluated models on confirmed weekly Influenza cases and confirmed weekly hospital admissions related to severe acute respiratory infections (SARI) in Germany. While the Influenza and SARI data is only available on a country level, the COVID-19 datasets are on a regional level (16 Bundesländer in Germany and 13 Régions in France, excluding overseas regions). Moreover, we also included the country-level data for the COVID-19 related datasets. In the SARI and Influenza datasets, the hospitalized and incident cases are provided, normalized to 100 thousand people (incidence), An overview of the used surveillance data can be found in Table 1. The German surveillance data were received from the Robert Koch Institute (RKI) (https://github.com/robert-koch-institut) and the French surveillance data from Santé Publique France (SPF) (https://www.data.gouv.fr/fr/organizations/sante-publique-france). For all models, the time series were log-transformed, because the raw data is locally expected to demonstrate exponential growth behavior. The daily data was smoothed using a centered moving average over seven days. For all models we used univariate time series, i.e. only past confirmed cases to forecast confirmed cases, and only past confirmed deaths to forecast confirmed

**Table 1. Surveillance data.** In total 8 different datasets were included in this study; confirmed COVID-19 cases, hospitalization and deaths for Germany and France, as well as confirmed Influenza cases and confirmed hospitalizations due to severe acute respiratory infections (SARI). The COVID-19 datasets are available on both country- and regional level, which results in 16 (regional) + 1 (country) time series for each of the German datasets and in 13 (region) + 1 (country) for each of the French datasets. All datasets were split into N training and evaluation windows. Details are explained in section 2.4.

| Name | Source | Period | N | Time Resolution | Spatial Resolution (# regions) | Training and Validation Data Dimensions |
|---|---|---|---|---|---|---|
| COVID Cases, Hosp., Deaths DE | RKI | 2020–2023 | 140 | daily | regional + country (16+1) | (70, 140) 14, 140) |
| COVID Cases, Hosp., Deaths FR | SPF | 2020–2023 | 140 | daily | regional + country (13+1) | (70, 140) (14, 140) |
| Influenza Cases DE | RKI | 2020 2024 | 30 | weekly | Country (1) | (52, 30) (2, 30) |
| SARI Hosp DE | RKI | 2014–2024 | 80 | weekly | Country (1) | (52, 80) (2, 80) |

deaths, etc. For model training and evaluation, the data was split into several training (fitting) and evaluation periods which is further detailed in Section 2.4.

## 2.2 Metadata

As metadata, we incorporated data from Google Trends following Wang et al. [24]. First, we identified the 20 top symptoms of COVID-19 which were used as search terms in Google Trends. By accessing the API (https://github.com/googleapis/google-api-python-client) we extracted the normalized daily number of counts each term was searched for. For smoothing we applied a centered moving average over seven days.

## 2.3 Base models

In the following, we introduce the used base models and explain why they are suitable for time series forecasting. The exact training and tuning procedure is explained in section 2.5.

**2.3.1 Linear regression.** Assuming that a pandemic follows exponential-like behavior—exponential growth and decay in waves, log-transforming the data will locally yield linear slopes. Therefore, linear regression can be used to fit linear models to the log-transformed data. Using the regression parameters, the fit can then be extrapolated to estimate short-term forecasts. We used the scikit-learn library (version 1.0.2) "linear-model".

**2.3.2 ARIMA.** Autoregressive integrated moving average (ARIMA) models use the statistical characteristics of stationary data. They are popular for time-series forecasting and have previously been applied to modeling of COVID-19 surveillance data [25–28]. A stationary series has no trends and consistently varies around its mean. That means short-term random time patterns can be extracted accordingly and used for forecasting. Here we employed a non-seasonal ARIMA model fitted to short-term periods which are not expected to show seasonal effects. This also applies to the Influenza and SARI data. Using seasonal ARIMA would only become effective when including at least two seasons. In this case, the ARIMA models depended on three parameters:

- $p$ the number of autoregressive terms

- $d$ the degree of differencing the data to make it stationary

- $q$ the number of lagged forecast errors

    With these parameters the general ARIMA forecasting equation is defined as:

$$\hat{y}_t = \mu + \varphi_1 y_{t-1} + \ldots + \varphi_p y_{t-p} - \theta_1 e_{t-1} - \ldots - \theta_q e_{t-q} \tag{1}$$

Here $\hat{y}$ corresponds to the forecast which is computed as the deviation of the mean $\eta$ of a stationary time series with $\varphi$, the slope parameters for each of the p previous values y, and q moving average parameters θ with autocorrelation errors e. This means the model learns to

predict future steps based on the mean of a stationary time series with adjusted autocorrelation errors and a lagged period [20].ARIMA employs the differencing technique to enhance the robustness with regard to the assumption of stationarity. Differencing refers to the process of computing the differences between consecutive values in a time series. Doing so transforms the time series to the fluctuations of consecutive values, which in first or second order often leads to stationarity [29]. To find the best parameters (*p*,*d*,*q*) we implemented the auto-ARIMA functionality which is part of the pmdarima library (version 2.0.3) [30]. This essentially corresponds to a hyperparameter tuning.

**2.3.3 Random Forest and XGBoost.** Both Random Forest and eXtreme Gradient Boosting (XGBoost) are based on decision trees. However, they differ to a great extent in their training algorithm. Random Forest builds an unweighted ensemble of decision trees, which are—by applying bagging—trained in parallel on different subsets of the data and then averaged [22]. In contrast, XGBoost builds its decision trees one after the other and corrects the residual errors made by the previously trained weighted decision tree ensemble using gradient descent [21]. Both models are commonly applied to tabular data but have also been shown to be successful in time series forecasting [31, 32], also for COVID-19 [33–36]. Since they are based on decision trees, they can only extrapolate based on previously seen training data. Therefore we applied the previously explained differencing technique [29], such that the difference between subsequent data points is forecasted, which is better represented in the training data. Afterwards the forecast was back-transformed accordingly and evaluated. For Random Forest we used the scikit-learn library (version 1.0.2) "ensemble" and for XGBoost the xgboost library (version 1.7.3).

**2.3.4 LSTM.** Recurrent Neural Networks (RNNs) are commonly used for sequencing data. Their advantage compared to standard neural networks is their internal memory, i.e., their ability to remember and learn the influence of previous steps on current steps. Opposed to standard RNNs, a Long Short Term Memory (LSTM) can learn longer-range time patterns of time series without suffering from the vanishing gradient problem [23]. LSTMs have also been applied for time series forecasting in COVID-19 [37, 38] Here we implemented an LSTM model in which the last hidden state—the state that contains the latent information about the time series—is decoded in a fully connected layer with output dimension according to the prediction window (14 days / 2 weeks). The LSTM model and fully connected layer were implemented using pytorch (version 1.11.0).

## 2.4 Sliding window approach for model training, tuning, and evaluation

We split the time series into training and evaluation windows for each region and dataset individually (see Table 1). For this purpose, we followed a sliding window approach (see Fig 1) with a training window size of 70 days for the data with daily resolution and 52 weeks for the data with weekly resolution, respectively. An evaluation window size of 14 days (daily data) / 2 weeks (weekly data), and a stride of 7 days (daily data) / 1 week (weekly data) were used. For the COVID-19 related datasets this resulted in 140 training and evaluation windows and for the Influenza and SARI datasets in 30 and 80 training and evaluation windows, respectively. The objective was to forecast the value of the time series 14 days ahead of time, counted from the end of the training window. ARIMA, Random Forest, as well as XGBoost, were trained on the whole training set, where each training sample corresponded to one sliding window. The log-linear regression was fitted on the last seven days (daily data) / five weeks (weekly data) of the training data. These models were applied separately for each region. With this we obtained the total number of training and evaluation data points (i.e. sliding windows) denoted in Table 1. Training and evaluation windows were strictly kept separate from each other to

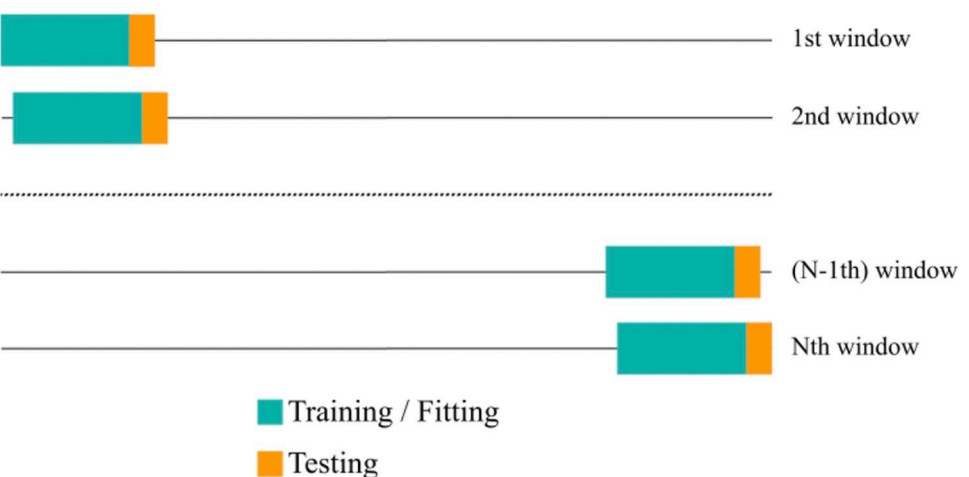

**Fig 1. Sliding window approach.** The time series was split into N training and evaluation windows. Models were tuned on the training windows using cross-validation (green), retrained with the best hyperparameters, and then forecasted 2 weeks ahead. Predictions were compared against observed values in the evaluation window (yellow).

prevent overoptimism, i.e. we trained on the first training period to evaluate on the first evaluation period, and then moved to the second training and evaluation periods.

For the LSTM model, however, we needed more data. Therefore, we applied a nested sliding window approach by creating training windows of size 7 days (daily data) / 5 weeks (weekly data) and evaluation windows of size 14 days (daily data) / 2 weeks (weekly data), with a stride of 1 day (daily data) / 1 week (weekly data) inside of the previously created outer sliding windows. Following Eq 2,

$$\#Sliding\ Windows = P - size_{ftw} + 1 \qquad (2)$$

where $P$ is the training period and $size_{ftw}$ is the training window size, this resulted in 63 sliding windows (daily data) / 48 sliding windows (weekly data) on the basis of 70 days (daily data) / 52 weeks (weekly data) training periods per training set. We decided to not train one LSTM model per region but to shuffle the regional windows (if available), to increase the amount of training data. Therefore, the LSTM's training objective was to predict 14 days (daily data) / 2 weeks (weekly data) ahead based on 7 days (daily data) / 5 weeks (weekly data) of training data. Thus, the dimension of the training sets changed, on the one hand because of the transformation of the period into sliding windows and on the other hand due to the aggregation of all regional datasets (only applicable for the COVID-19 datasets, in which regional data was available). For the German COVID-19 datasets the training set dimensions were (17*63, 140) and for the French COVID-19 datasets (13*63, 140), and for the Influenza and SARI datasets (48, 30) and (48, 80). Note, here the first dimension corresponds to the number of sliding windows and is not in unit days or weeks, but each window has length 7 days (daily data) or 5 weeks (weekly data). Still the LSTM was trained and evaluated separately for each surveillance data, i.e. training on confirmed cases to forecast confirmed cases.

We then tuned the hyperparameters of Random Forest, XGBoost, and LSTM models for each training window and region if applicable using Optuna (version 2.10.1). For more information about hyperparameter tuning, we refer to the (see S1 Table). Since we were using the auto–ARIMA functionality, the hyperparameter tuning was done via a grid search, where the maximum parameter values for $(p,q)$ were set to (14,14). We split the training windows into 80% training and 20% validation for hyperparameter tuning. After hyperparameter tuning we

retrained the models—using the best hyperparameters for each fitting window—on the whole training data and progressively predicted and evaluated on the evaluation windows.

We additionally wanted to evaluate the models' performances on the COVID-19 winter season of 2023–2024 for an evaluation on the most recent wave, and to test whether the models are still applicable even after COVID-19 has become more and more endemic (according to the RKI). Data was only available in Germany, in France incident cases, deaths and hospitalization recording was stopped end of June 2023 (see https://www.data.gouv.fr/fr/organizations/sante-publique-france/#/datasets). Additionally, we decided to only work with incident confirmed cases and hospitalization, since the number of recorded COVID-19 deaths in Germany stayed below 100 counts throughout this winter season. With these extended datasets up to the end of May 2024, we created 208 training and evaluation windows.

## 2.5 Model evaluation metrics

To evaluate the performance of the base models and later the ensemble we used the mean absolute percentage error (MAPE) as a metric:

$$MAPE = \frac{1}{n}\sum_{i=1}^{n}|\frac{Y_i - \hat{Y}_i}{Y_i}|*100 \tag{3}$$

with $Y$ as the observed value, $\hat{Y}$ as the predicted value, and $n$ as the number of data points, in our case the prediction window (14 days / 2 weeks). The MAPE represents the deviation of the prediction from the observed data in percent and is therefore a more tangible measure than the mean squared error. The MAPE alone should not be used for determining the performance of a model, since it is scale-dependent [39]. However, it is a good measure to quantitatively compare the performances of different models.

## 2.6 Baseline ensemble approaches

As a baseline, we implemented two basic ensemble algorithms, more specifically the mean and the median of the model forecasts. Additionally, we built an ensemble algorithm that always chooses the model that performed best in the previous evaluation period (Prev.-Best). This corresponds to a first step in accounting for the dynamics of the pandemic and thus the dynamic performance of each base model.

## 2.7 Dynamic model selection and stacking

We here propose two possible extensions of the baseline ensemble methods discussed before: i) dynamic model selection and ii.) dynamic model stacking, an extension of a classical stacked regressor approach [40]. In practice, we realize both approaches by training a meta-model, which we chose as a simple MLP architecture with a tunable hidden layer size. The input for the meta-model constituted of the predicted values as well as estimated prediction performances of all base models, by concatenating the MAPEs of the previous evaluation period to the log-transformed forecasts of the current evaluation period. Therefore, the MLP has five input vectors—one per base model. After each hidden layer a rectified linear unit (ReLU) activation function is applied. The output layer is designed to hold one node per model including a softmax head at the end. As mentioned above, there are two learning objectives:

1. Dynamic model selection: The meta-model is trained to always select the model with the highest softmax output. This essentially corresponds to a classification task, where the model with the highest probability score is selected.

2. Dynamic model stacking: The meta-model is trained to multiply the base model forecasts by the individual softmax inputs. These weighted outputs are then aggregated into one final ensemble output. This essentially corresponds to a weighted mean, since the softmax outputs add up to 1.

Both learning objectives are trained with a weighted MSE (WMSE) loss function:

$$WMSE = \frac{1}{n}\sum\nolimits_{i=1}^{n} \frac{(Y_i - \hat{Y}_i)^2}{Y_i} \tag{4}$$

with $Y$ as the observed value, $\hat{Y}$ as the predicted value, and $n$ as the number of data points, in our case the prediction window (14 days / 2 weeks). The WMSE has the advantage that it penalizes the relative deviation rather than the absolute deviation. For example: say the observed value was 100 and the predicted value was 101 the MSE would be 1. It would be the same for the observed value being 1 and the predicted value being 2. However, the relative deviation would be 1% vs. 100%. Weighting the MSE by the observed values results in normalizing this error to the observed value scale. In this case, the WMSE would be 0.01 and 1, respectively—the deviation of 100% is accordingly penalized much more than the deviation of 1%.

## 2.8 Inclusion of metadata

Our above-described modeling approaches used only surveillance data, their forecasts, and estimated prediction performances as input for the base models and meta-model, respectively. In our previous publication, we showed that social media data is not only correlated with surveillance data but can also be used to forecast up- and downtrends of pandemic waves [24]. Therefore, we wanted to test if the inclusion of social media data or further metadata could improve the prediction performance of the meta-model. We employed Google Trends data as described above and applied a sliding window approach, where we used the past $n$ ($n$ = 2,3,4) weeks (before the last date of our fitting period) as the training period for the metadata. To extract time patterns, we used an LSTM model and concatenated the last hidden state to the input of the meta-model, extending the input feature vector to include the forecasted value, the prediction performance estimated from the previous evaluation period, and now the information coming from the metadata. The meta-model was then trained in the same way as before, but now the weights of the LSTM were also updated according to the weighted MSE loss between the output and the observed values. Due to the high computational burden and to be consistent with Wang et al. [24], we evaluated this approach on German surveillance data only.

## 2.9 Overall ensemble model pipeline

The final ensemble model pipeline can be seen in Fig 2. The surveillance data, which was previously split into training and evaluation windows according to the sliding window approach explained above, is used as input for the base models. The tuned base models are trained in parallel and create a rolling forecast based on the evaluation windows. After each evaluation window, the baseline models (mean, median, Prev.-Best) are created. The predictions are evaluated using the current validation data and the MAPE as a metric. The base models' forecasts together with their performance on the previous evaluation period are concatenated to form the input vectors of the meta-model. If metadata is included the metadata is fed into an LSTM model, of which the last hidden state—the latent representation of the metadata—is concatenated to the input vectors of the meta-model. As described in Section 2.4 the overall data was split into 80% training and 20% test. The training data was further split into 5 folds

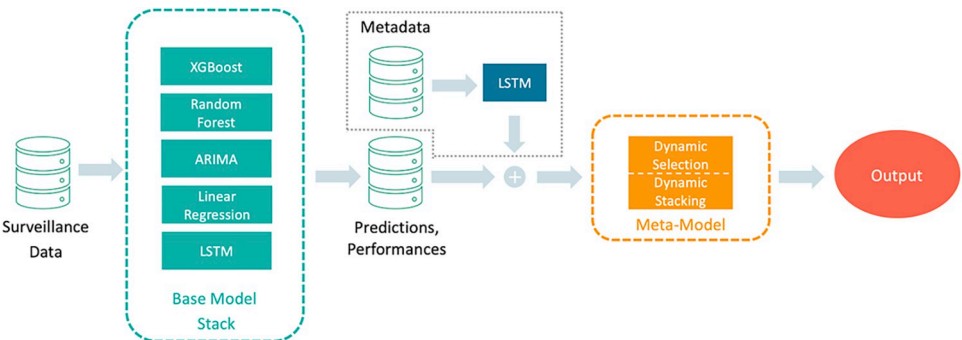

**Fig 2. Overall ensemble model pipeline.** The surveillance data is fed into the base models which produce forecasts. All forecasts and their evaluation (plus the latent representation of the metadata) are used as input for the meta-model which outputs forecasts either based on dynamic selection or dynamic stacking.

for an inner 5-fold cross-validation and hyperparameter tuning (see S1 Table). Finally, the models' performances over the test data were averaged and an output containing these mean performances was returned. The code for the ensemble model can be accessed on Git Hub (https://github.com/SCAI-BIO/Dynamic_Ensemble).

## 2.10 Model ranking and post-hoc analysis

To quantitatively compare the model performances across datasets we first ranked all models according to a consensus ranking [41]—allowing for ties—based on Kemeny's axiomatic approach [42]. The algorithm compares models pairwise and counts how often one model is ranked above the other. The total sum of counts is then used to form the consensus ranking. This ranking alone, however, does not necessarily mean that one model's performance is significantly different from another model. To test for statistical significance across models we thus used a Kruskal-Wallis test [43]. To test which individual models differed significantly from each other we then used a pairwise Wilcoxon test as post-hoc test [44]. All p-values are adjusted for multiple testing based on the Holm-Bonferroni method [45]. Statistical tests were implemented using R (version 4.3.0) and the libraries ConsRank (version 2.1.4) and stats (version 4.3.0).

## 3. Results

In the following, we display the results on the country level and the regional results aggregated (mean over all regional results) at the country level. The complete set of results can be found in the S1 Results. Model performances are displayed as the mean MAPE of all test windows in percent together with its standard error in parentheses. We use the following abbreviations for the models: Linear Regression—LR, XGBoost- XG, Random Forest—RF, and ensemble baseline by best model selection—Prev.-Best. Since the Influenza dataset only contains 30 test windows (and just 6 test windows for the meta-model) the results should be interpreted cautiously, as the small sample size leads to a reduced statistical meaningfulness. Additionally, we provide the results from the consensus ranking and the Kruskal-Wallis as well as the Wilcoxon test. For all results, the Kruskal-Wallis test returned a significant p-value of less than 5%.

## 3.1 Base models versus baseline ensembles

First, we evaluated the base and baseline ensemble models by computing and testing a rolling forecast over the full time series. This resulted in 140 test windows for the daily COVID-19 datasets, 30 test windows for the weekly Influenza cases, and 80 test windows for the weekly

SARI hospitalization. The results are summarized in Table 2. For a better overview, we colored the three best models for each dataset / dataset aggregation. First, looking at the base models' performances on the daily COVID-19 datasets, it can be seen that mostly linear regression and ARIMA performed best and LSTM and XGBoost worst. On the weekly dataset, Random Forest and XGBoost were able to perform similarly well as ARIMA. Here the linear regression showed a reduced performance. Taking a look at the baseline ensemble methods shows that mean and median baseline ensembles rarely performed as one of the three best models, but the

**Table 2. Base models versus baseline ensemble methods.** The performances are given as the mean MAPE and its standard error in parentheses of the N test windows for each dataset / dataset aggregation. The best three models are colored according to the provided legend. DE (FR) stands for German (France) country level and DE_reg (FR_reg) for German (France) regional level aggregated to country level. The last two rows display the results from the consensus ranking over all datasets and models.

| Geography | LR | LSTM | XG | RF | ARIMA | Mean | Median | Prev.-Best |
|---|---|---|---|---|---|---|---|---|
| **Daily COVID-19 Cases DE (N = 140)** | | | | | | | | |
| DE | 21.07 (2.27) | 38.10 (3.76) | 27.90 (1.71) | 24.54 (1.36) | 19.15 (1.14) | 23.04 (1.30) | 22.51 (1.15) | 17.37 (1.04) |
| DE_reg | 29.84 (2.61) | 38.56 (7.44) | 31.43 (1.40) | 28.22 (1.08) | 24.93 (1.08) | 27.27 (1.88) | 26.26 (1.05) | 24.75 (1.05) |
| **Daily COVID-19 Hospitalization DE (N = 140)** | | | | | | | | |
| DE | 12.75 (0.77) | 28.95 (2.21) | 20.51 (1.39) | 19.17 (1.15) | 8.88 (0.66) | 14.97 (0.90) | 15.40 (0.98) | 11.94 (1.54) |
| DE_reg | 27.28 (1.65) | 27.06 (2.46) | 28.22 (1.29) | 25.79 (1.01) | 17.64 (0.72) | 21.39 (0.92) | 21.30 (0.75) | 20.19 (0.87) |
| **Daily COVID-19 Deaths DE (N = 140)** | | | | | | | | |
| DE | 19.85 (1.27) | 37.55 (2.04) | 27.29 (1.72) | 24.84 (1.39) | 21.17 (1.02) | 22.31 (1.15) | 23.00 (1.19) | 20.10 (1.28) |
| DE_reg | 37.41 (0.95) | 26.88 (2.20) | 29.93 (1.36) | 27.13 (0.96) | 25.35 (0.87) | 28.21 (0.73) | 30.94 (0.92) | 24.52 (1.18) |
| **Daily COVID-19 Cases FR (N = 140)** | | | | | | | | |
| FR | 20.71 (1.33) | 45.94 (4.46) | 38.08 (3.44) | 33.89 (2.71) | 22.67 (1.89) | 28.63 (2.28) | 29.50 (2.42) | 23.01 (2.08) |
| FR_reg | 23.47 (1.27) | 37.54 (4.32) | 41.14 (3.97) | 37.04 (3.05) | 25.13 (2.05) | 29.75 (2.21) | 28.10 (1.84) | 24.07 (1.43) |
| **Daily COVID-19 Hospitalization FR (N = 140)** | | | | | | | | |
| FR | 16.45 (1.01) | 33.61 (1.94) | 25.95 (1.66) | 24.26 (1.43) | 16.84 (1.05) | 20.43 (1.19) | 21.02 (1.22) | 17.57 (1.11) |
| FR_reg | 26.90 (1.19) | 29.39 (1.37) | 28.87 (1.45) | 26.52 (1.17) | 23.08 (0.91) | 23.83 (0.97) | 24.70 (1.02) | 23.90 (0.92) |
| **Daily COVID-19 Deaths FR (N = 140)** | | | | | | | | |
| FR | 16.77 (1.35) | 30.63 (1.87) | 20.79 (1.27) | 19.32 (1.19) | 17.09 (1.12) | 17.07 (1.03) | 17.96 (1.10) | 16.70 (1.42) |
| FR_reg | 32.40 (1.17) | 22.7 (1.23) | 25.54 (0.87) | 23.34 (0.67) | 21.42 (0.62) | 23.22 (0.70) | 24.91 (0.77) | 22.48 (0.79) |
| **Weekly Influenza Cases DE (N = 30)** | | | | | | | | |
| DE | 47.57 (3.67) | 23.53 (10.18) | 14.85 (4.95) | 15.44 (4.70) | 11.28 (3.54) | 38.67 (2.31) | 44.34 (2.32) | 28.46 (10.57) |
| **Weekly SARI Hospitalization DE (N = 80)** | | | | | | | | |
| DE | 16.41 (1.43) | 20.07 (1.96) | 12.62 (1.27) | 12.92 (1.28) | 12.90 (1.34) | 12.36 (1.14) | 13.05 (1.16) | 15.64 (1.60) |
| **Consensus Ranking** | | | | | | | | |
| All | 6 | 4 | 7 | 5 | 1 | 2 | 3 | 1 |
| | | | Best Model | 2nd Best Model | | 3rd Best Model | | |

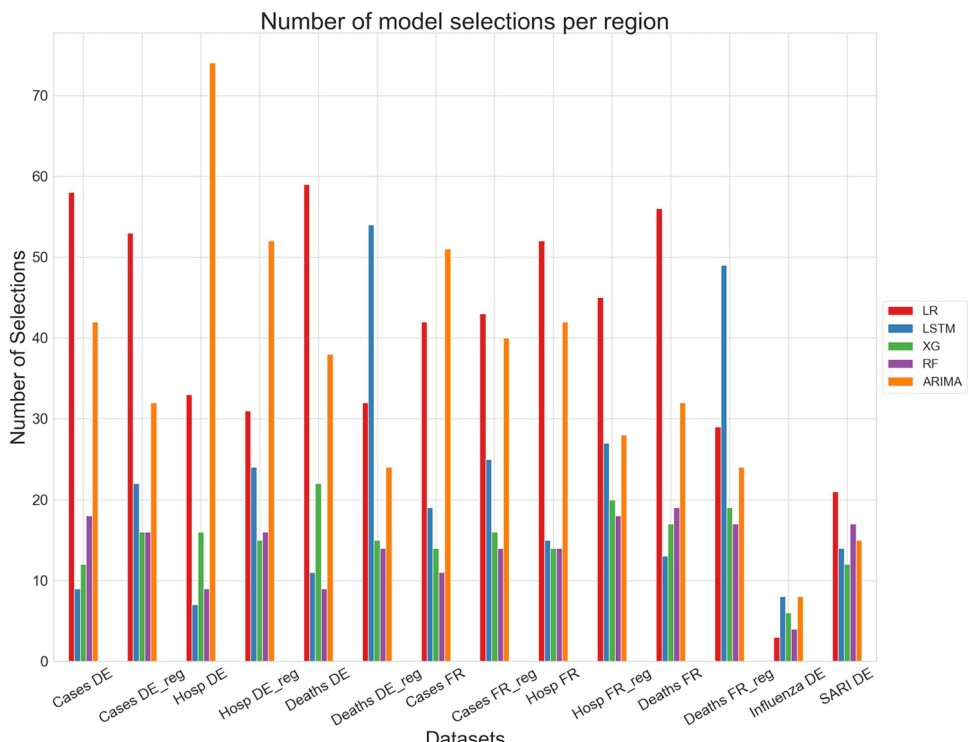

**Fig 3. Number of model selections per region bar plot.** DE (FR) stands for German (France) country level and DE_reg (FR_reg) for German (France) regional level aggregated to country level.

Prev.-Best method was able to outperform most of the base models in many instances; at least for the daily COVID-19 datasets. Evaluated on the SARI hospitalization dataset, mean managed to be the best model. Since Prev.-Best always selects the best model of the previous week, we integrated a counter to keep track of this selection. The number of model selections per dataset can be seen in Fig 3. This agrees with the results displayed in Table 2. The base models that performed best were also the ones being selected most often. But, still, the other base models were selected a considerable amount of times. Finally, it can be observed that the variance of the Prev.-Best method performances tended to be smaller than the variance of the base models (perhaps excluding ARIMA) performances. Now looking at the consensus ranking, we can see that ARIMA and Prev.-Best were both ranked first, followed by the mean and median baseline ensemble methods. Table 3 shows the p-values computed by the pairwise Wilcoxon test. It can be seen that indeed no significant difference between the Prev.-Best method and

**Table 3. Base models versus baseline ensemble approaches: Pairwise Wilcoxon test (multiple testing adjusted p-values).** Statistically significant tests (p-value < 0.05) are shown in bold.

|            | ARIMA    | LR       | LSTM     | Mean     | Median   | Prev.-Best | RF       |
|------------|----------|----------|----------|----------|----------|------------|----------|
| **LR**     | **<0.001** | -        | -        | -        | -        | -          | -        |
| **LSTM**   | **<0.001** | 0.070    | -        | -        | -        | -          | -        |
| **Mean**   | **<0.001** | **<0.001** | **<0.001** | -        | -        | -          | -        |
| **Median** | **<0.001** | **0.008** | 0.889    | **<0.001** | -        | -          | -        |
| **Prev. Best** | 0.842 | **<0.001** | **<0.001** | **<0.001** | **<0.001** | -        | -        |
| **RF**     | **<0.001** | 0.889    | 0.201    | **<0.001** | 0.094    | **<0.001** | -        |
| **XG**     | **<0.001** | **<0.011** | **<0.001** | **<0.001** | **<0.001** | **<0.001** | **<0.001** |

**Table 4. Comparison of ensemble modeling approaches.** The performances are given as the mean MAPE and its standard error in parentheses of the N test windows for each dataset / dataset aggregation. DE (FR) stands for German (France) country level and DE_reg (FR_reg) for German (France) regional level aggregated to country level.

| Geography | Prev.-Best | Dynamic Model Selection | Dynamic Model Stacking |
|---|---|---|---|
| Daily COVID-19 Cases DE (N = 28) | | | |
| DE | 17.07 (2.38) | 28.94 (7.10) | 24.58 (2.41) |
| DE_reg | 22.50 (2.13) | 30.44 (3.59) | 26.29 (2.37) |
| Daily COVID-19 Hospitalization DE (N = 28) | | | |
| DE | 12.25 (2.01) | 24.51 (3.38) | 14.32 (2.01) |
| DE_reg | 17.47 (1.36) | 24.39 (2.16) | 19.68 (2.05) |
| Daily COVID-19 Deaths DE (N = 28) | | | |
| DE | 24.87 (3.83) | 27.27 (4.57) | 21.99 (2.40) |
| DE_reg | 31.71 (1.78) | 34.59 (8.26) | 22.86 (2.70) |
| Daily COVID-19 Cases FR (N = 28) | | | |
| FR | 21.53 (3.08) | 31.36 (8.19) | 20.48 (2.58) |
| FR_reg | 22.80 (2.90) | 28.79 (5.23) | 22.09 (2.46) |
| Daily COVID-19 Hospitalization FR (N = 28) | | | |
| FR | 21.35 (2.76) | 29.80 (5.04) | 17.60 (2.62) |
| FR_reg | 28.88 (2.09) | 32.53 (4.03) | 20.73 (1.82) |
| Daily COVID-19 Deaths FR (N = 28) | | | |
| FR | 19.32 (3.83) | 21.06 (4.57) | 14.07 (2.40) |
| FR_reg | 26.76 (1.78) | 24.41 (8.26) | 17.99 (2.70) |
| Weekly Influenza Cases DE (N = 6) | | | |
| DE | 10.87 (7.76) | 35.80 (8.17) | 7.82 (3.82) |
| Weekly SARI Hospitalization DE (N = 16) | | | |
| DE | 15.15 (3.66) | 17.2 (2.64) | 13.19 (3.12) |
| Consensus Ranking | | | |
| All | 2 | 3 | 1 |

ARIMA could be found. However, they were both found to be significantly different than all other methods.

## 3.2 Baseline ensembles versus dynamic model stacking and selection

Next, we evaluated our proposed Dynamic Model Stacking and Selection approaches against the previously tested Prev.-Best method. Since the meta-model was trained on 80% of the test windows, the number of test windows for the meta-model was reduced to 28 for the daily COVID-19 dataset, 6 for the weekly Influenza cases, and 16 for the weekly SARI hospitalization. According to the results presented in Table 4 Dynamic Selection was not able to outperform Prev.-Best. However, Dynamic Model Stacking outperformed Prev-Best and Dynamic Model Selection on the French and German COVID-19 deaths datasets and was the second-best model on the German COVID-19 hospitalization dataset. Moreover, it outperformed Dynamic Model Selection and Prev.-Best on the weekly datasets. Also, the variance of the Dynamic Model Stacking approach tended to be reduced compared to the other methods. The results of the consensus ranking are again in line with the findings above. The Dynamic Model Stacking method is ranked first being significantly better than Prev.-Best (see Table 5).

For the sake of completeness, we also list in Table G in S1 Results an additional comparison of Dynamic Model Stacking against all base models, which have been re-trained on the same dataset. Also, in this comparison, Dynamic Model Stacking could significantly outperform ARIMA as the best base model (p = 5.74E-3).

**Table 5. -best versus dynamic model stacking and selection: Pairwise Wilcoxon test (adjusted p-values).** **Prev.** Statistically significant tests (p-value < 0.05) are shown in bold.

| | Prev.-Best | Dynamic Selection | Dynamic Model Stacking |
|---|---|---|---|
| **Dynamic Selection** | 0.324 | - | - |
| **Dynamic Model Stacking** | **<0.001** | **<0.001** | - |

## 3.3 Potential benefits of including metadata

Additionally, we wanted to assess whether the inclusion of metadata—here Google Trends symptom counts—could further enhance our proposed Dynamic Model Stacking approach. We could not find an improvement by including metadata. Both Dynamic Model Stacking with and without metadata were ranked on position 1 after consensus ranking and the Wilcoxon-test showed no significant differences between both approaches (Tables O-Q in S1 Results)

## 3.4 Model evaluation including 2023–2024 winter season

Finally, models were evaluated also on the 2023–2024 winter season. For this purpose, base models were first trained and tested on 208 windows of the extended datasets of COVID-19 incident cases and hospitalization in Germany. The performance of the base models and baseline ensemble methods were similar to the previous results (see Table R in S1 Results). Also, here ARIMA and Prev.-Best were ranked first, followed by mean and median consensus ranking. The train-test split of the meta-model resulted in 42 evaluation windows over the period of August 2023 to June 2024 covering the whole winter season 2023–2024. According to consensus ranking, Dynamic Model Stacking was found to outperform ARIMA and Prev.-Best, as before (see Table 6).

## 3.5 Model performance is robust over time

When plotting the test set performance of our proposed Dynamic Model Stacking as a function of time, no significant increase in model prediction errors for the hospitalization rate could be observed (Fig 4). Similarly, no significant trend was found in the validation window performances of ARIMA between 2020 and 2024 (Fig A in S1 Results). Hence, we conclude that our proposed short-term forecasting ensemble approach is rather robust against significant changes of the epidemic process as well as exogenous influence factors during that time.

**Table 6. Comparison of ensemble modeling approaches and ARIMA.** The performances are given as the mean MAPE and its standard error in parentheses of the N test windows of the 2023–2024 winter season for each dataset / dataset aggregation. DE stands for German country level and DE_reg for German regional level aggregated to country level. The full table can be seen in the appendix Table S in S1 Results.

| Geography | ARIMA | Mean | Median | Prev.-Best | Dynamic Model Stacking |
|---|---|---|---|---|---|
| **Daily COVID-19 Cases DE (N = 42)** | | | | | |
| DE | 19.51 (3.19) | 16.96 (2.75) | 18.27 (3.12) | 18.35 (2.63) | 18.36 (2.42) |
| DE_reg | 27.36 (4.37) | 27.53 (6.59) | 26.89 (4.30) | 25.40 (4.09) | 23.91 (3.08) |
| **Daily COVID-19 Hospitalization DE (N = 42)** | | | | | |
| DE | 13.64 (1.98) | 14.86 (2.15) | 14.61 (2.19) | 15.40 (2.15) | 16.48 (2.28) |
| DE_reg | 22.30 (3.36) | 26.11 (4.72) | 24.45 (3.39) | 24.11 (3.93) | 20.49 (2.46) |
| **Consensus Ranking** | | | | | |
| All | 2 | 3 | 4 | 2 | 1 |

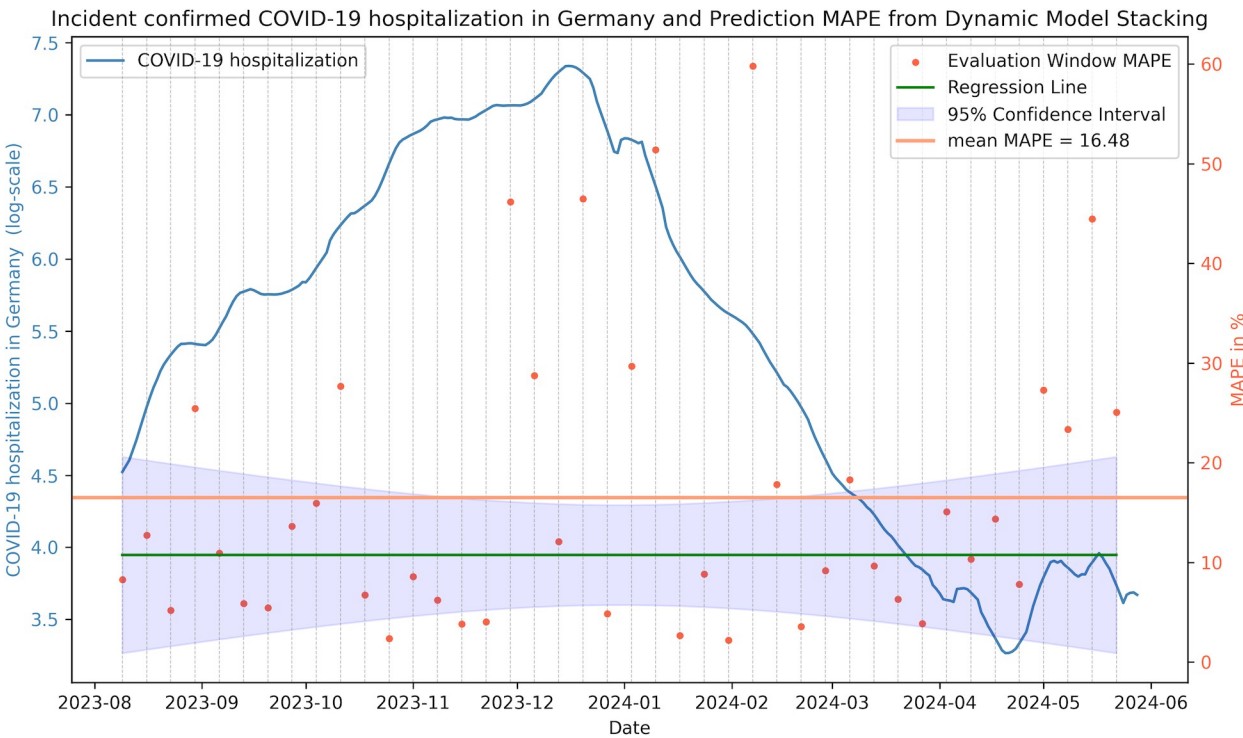

**Fig 4. Dynamic Model Stacking performance on the test set.** The blue line shows the COVID-19 hospitalization in Germany. The orange dots denote the model performance (MAPE) in % of the evaluation window 7 days before and 7 days after (14 days ahead prediction). The mean MAPE given as orange horizontal. Additionally, a robust linear regression fit including a 95% Confidence Interval was added to show the trend of the model performance.

## 4. Discussion

The COVID-19 pandemic highlighted the need for robust models that can accurately forecast the spread of the pandemic and can adjust dynamically to external factors such as newly imposed non-pharmaceutical interventions, new virus variants, vaccination, seasonal effects, and others. In this work, we initially tested and compared different autoregressive machine learning models against basic ensemble methods (mean, median, Prev.-Best). This demonstrated no statistically significant benefit of simple ensemble techniques compared to a state-of-the-art ARIMA time series forecasting model. Only Prev.-Best was found to perform en par with ARIMA. Even though the other base models did not perform as well as ARIMA we still found them to be frequently selected as significantly often as being the best model in the previous test window. Therefore, we decided to keep these other base models in the model ensemble.

We then developed a meta-model that can either dynamically select one of the base model predictions or add the weighted base model predictions into one joint forecast. Interestingly, only Dynamic Model Stacking turned out to outperform Prev.-Best while at the same time showing a reduced variance in prediction performance. The inclusion of Google Trends symptom counts as metadata could not further improve Dynamic Model Stacking significantly and thus cannot be recommended. We believe that metadata–such as Google Trends–could potentially improve model forecasts in other scenarios, since Google Trends data has been found to provide an early indicator of trends seen later on in surveillance data [24]. This aspect should be further investigated in future research.

Additional evaluation of our proposed Dynamic Model Stacking on data of the 2023/2024 winter season confirmed the robustness of our approach. Furthermore, we found test set performances to remain stable over the entire COVID-19 pandemic.

Comparing the performances on the country and regional level we found Dynamic Model Stacking on the country level to be superior. We assume that a possible reason could be the data quality of the regional data. Since we are working with surveillance data that needs to be registered at local health departments, it can happen that mistakes are made on a regional level. These mistakes would have reduced effects when the regional data is aggregated to the country-level data.

In general, we saw considerable differences between performances of models trained on data with a daily versus those trained on a weekly time resolution. Specifically, the SARI hospitalization data suggests that decision tree-based models—especially Random Forest—and the mean and median ensemble methods work well here. In this regard, we should point out that in the weekly data, the task is just to forecast the next two data points (2 weeks) which seems to be handled well by decision tree models. Linear regression struggles here because the model is fit to the past 5 data points (i.e. weeks), hence resulting in over-smoothing. Due to the low time resolution in the weekly datasets, base models may be overfitted and thus fail to predict subsequent test periods. In contrast, this effect is significantly reduced when training on data with a daily time resolution. We believe that this explains why the ensemble baseline model, Prev.-Best, which was among the top-performing models when trained on the daily data, whereas it showed significantly poorer performance when trained on the weekly data.

We should mention the limitations of the non-COVID datasets, specifically limited sample size and, in case of the Influenza, also seasonal fluctuations (see S1 Fig). Moreover, the SARI dataset contains hospitalization due to different pathogens. Hence, there are different epidemic processes and dynamics compared to COVID-19. Testing our modeling approaches on these data thus provides insights into their principal robustness against differences in epidemic processes.

A comparison of our findings with those in other studies is challenging because different datasets (perhaps even just one wave rather than a whole pandemic), different forecasting horizons, and different metrics have been used. Paireau et al. [11] developed an ensemble model (mean) and forecasted among other indicators the COVID-19 hospitalization in France. On country-level data, they documented a mean MAPE of 20% and on aggregated regional level of 30% for a 14-day forecast horizon. Our best model—evaluated on the French COVID-19 hospitalization dataset—achieved a mean MAPE of around 17% and around 20% to 23% (for the meta-model test windows or all test windows) on country-level and regional aggregated country-level data, respectively. Heredia Cacha et al. [10] forecasted COVID-19 cases in Spain using different ensemble methods (mean, median, weighted average) and documented a mean MAPE of around 30% for a 14-day forecasting horizon. We achieved a MAPE of around 17% to 25% for forecasting the number of COVID-19 cases in Germany and France. Stating that our models are better than the ones of Paireau et al. and Heredia Cacha et al. would not be fair, though, since we are not using the exact same data. However, this comparison confirms that our models are generally competitive with others reported in the literature.

## 5. Conclusion

A major challenge for the modeling of pandemic situations, specifically COVID-19, is their highly dynamic character. Rapid introduction of non-pharmaceutical interventions, newly emerging virus variants, vaccinations, and seasonal effects strongly violate the typical assumption of stationarity in time series modeling and forecasting and thus negatively affect the generalization ability of models. In this regard, we here proposed a novel ensemble learning

strategy, in which a meta-model learns to dynamically weigh and integrate a set of base models based on currently observed data and past performance indicators. Based on results from 8 datasets, our Dynamic Model Stacking approach was able to outperform state-of-the art time series forecasting techniques, such as ARIMA, and other ensemble learning approaches. Furthermore, we could show that our method could not be further improved by adding further metadata, such as Google searches. While in this work we focused on autoregressive base models, our proposed Dynamic Model Stacking algorithm should in principle be applicable to any type of base modeling approach, including mechanistic models.

In this study we focused on short-term forecasting of the observed, confirmed number of incident cases, deaths and hospitalizations as well as observed, confirmed Influenza cases and SARI hospitalization. Due to underreporting, including asymptomatic infections, no available tests or specific test strategies, and decentral reporting these observations may differ from reality. However, since we focused on forecasts over short time periods only, these reporting biases have likely only a minor effect–especially in smooth periods.

The reporting of confirmed cases leads to the more general question of data quality. In Germany confirmed cases were reported by local health authorities to the RKI. However, measurement errors and underreporting might differ between those local health authorities and thus lead to discrepancies in data quality.

The definition of death due to COVID-19 can vary [46]. In the individual case it can be extremely challenging to identify a prior COVID-19 infection as the cause for death. Moreover, it is unlikely that all deaths during the pandemic were tested for a prior COVID-19 infection.

Similarly, hospitalization rates due to COVID-19 infection might suffer from a certain level of underreporting. However, we believe that the underreporting rate here was much lower than for confirmed cases. Therefore, we think that the better performance of the models in forecasting COVID-19 hospitalization could be due to higher data quality.

Of course, our proposed Dynamic Model Stacking approach is not without limitations. Most importantly, machine learning and statistical methods need a sufficient amount of training data, i.e. retrospective pandemic data, which are not always available at the beginning of a pandemic. A potential strategy in future pandemics might thus be to start building a collection of comparably simple base models, specifically including ARIMA and possibly also mechanistic models, and then to train Dynamic Model Stacking once sufficient historical data is available. We here only tested autoregressive base models, which may fail at critical turning points of the disease dynamics. Hence, in the future the inclusion of well calibrated mechanistic base models to our ensemble could be a recommendable strategy. Opposed to statistical and machine learning models, mechanistic models offer the possibility of counterfactual simulations for scenario planning. Furthermore, mechanistic models are often thought to be better understandable than machine learning models due to their explicit mathematical nature.

While we only evaluated Dynamic Model Stacking on surveillance data of COVID-19, SARI, and Influenza, our method is not limited per se to these data. Dynamic Model Stacking could potentially be applied also to other areas, where non-stationary time series forecasting plays a role, e.g. traffic, energy consumption, air flights, and others. Moreover, future work could apply Dynamic Model Stacking to age-distributed data, especially to the vulnerable, mostly elderly population.

## Supporting information

**S1 Table. Hyperparameters for base models and meta-models.**
(PDF)

**S1 Fig. Comparison SARI and Influenza.** SARI and Influenza incidence from May 2022 to May 2023.
(TIFF)

**S1 Results. Complete results.**
(DOCX)

## Author Contributions

**Conceptualization:** Holger Fröhlich.

**Data curation:** Jonas Botz, Diego Valderrama, Jannis Guski, Holger Fröhlich.

**Formal analysis:** Jonas Botz, Diego Valderrama, Jannis Guski, Holger Fröhlich.

**Funding acquisition:** Holger Fröhlich.

**Investigation:** Jonas Botz, Diego Valderrama, Jannis Guski, Holger Fröhlich.

**Methodology:** Holger Fröhlich.

**Project administration:** Holger Fröhlich.

**Resources:** Holger Fröhlich.

**Supervision:** Holger Fröhlich.

**Validation:** Jonas Botz, Diego Valderrama, Jannis Guski, Holger Fröhlich.

**Visualization:** Jonas Botz, Diego Valderrama, Jannis Guski, Holger Fröhlich.

**Writing – original draft:** Jonas Botz, Diego Valderrama, Jannis Guski, Holger Fröhlich.

**Writing – review & editing:** Jonas Botz, Diego Valderrama, Jannis Guski, Holger Fröhlich.

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
