## [Decision Letter · Decision Letter 0]

5 Jun 2024

PGPH-D-24-00498

A dynamic ensemble model for short-term forecasting in pandemic situations

Dear Dr. Botz,

Thank you for submitting your manuscript to PLOS Global Public Health. After careful consideration, we feel that it has merit but does not fully meet PLOS Global Public Health’s publication criteria as it currently stands. Therefore, we invite you to submit a revised version of the manuscript that addresses the points raised during the review process.

In particular, the idea of using AR models to predict the course of an emerging disease such as COVID-19 is problematic. These models, by their nature, cannot adequately capture the nonstationary dynamics typical of pandemics. It is critical that you reformulate the article to clearly highlight these limitations, positioning AR models as tools for short-term predictions over stable periods, not as solutions for critical moments in a pandemic. In addition, the practical issues raised by the reviewers regarding underreporting are crucial. The COVID-19 pandemic has seen significant changes in case reporting practices, which can substantially affect the performance of your model. This issue needs to be addressed in depth.

We look forward to receiving your revised manuscript.

Kind regards,

Francesco Branda, Ph.D.

Academic Editor

Journal Requirements:

1. Please note that PLOS Global Public Health has specific guidelines on code sharing for submissions in which author-generated code underpins the findings in the manuscript. In these cases, all author-generated code must be made available without restrictions upon publication of the work. Please review our guidelines at https://journals.plos.org/plosglobalpublichealth/s/materials-and-software-sharing#loc-sharing-code and ensure that your code is shared in a way that follows best practice and facilitates reproducibility and reuse.

2. In your Methods section, please include additional information about your dataset and ensure that you have included a statement specifying whether the collection and analysis method complied with the terms and conditions for the source of the data.

3. Please amend your detailed Financial Disclosure statement. This is published with the article. It must therefore be completed in full sentences and contain the exact wording you wish to be published.

4. We ask that a manuscript source file is provided at Revision. Please upload your manuscript file as a .doc, .docx, .rtf or .tex.

Additional Editor Comments (if provided):

Reviewers' comments:

Reviewer's Responses to Questions

**Comments to the Author**

1. Does this manuscript meet PLOS Global Public Health’s publication criteria? Is the manuscript technically sound, and do the data support the conclusions? The manuscript must describe methodologically and ethically rigorous research with conclusions that are appropriately drawn based on the data presented.

Reviewer #1: Partly

Reviewer #2: Yes

2. Has the statistical analysis been performed appropriately and rigorously?

Reviewer #1: No

Reviewer #2: Yes

3. Have the authors made all data underlying the findings in their manuscript fully available (please refer to the Data Availability Statement at the start of the manuscript PDF file)?

Reviewer #1: Yes

Reviewer #2: Yes

4. Is the manuscript presented in an intelligible fashion and written in standard English?

Reviewer #1: Yes

Reviewer #2: Yes

5. Review Comments to the Author

Reviewer #1: The article shows the performance of the autoregressive ML models, broadly understood, including AR, MVAR (later abbey AR ML models), in a task of predicting the course of COVID-19 pandemic in years 2020-2022. It also reports the performance of AR ML models for two other diseases INFLUENZA an SARI (also in non-pandemic years 2014-2024.)

### Major Comment:

In accordance with the guidelines for reviewers, the Journal is open to presenting papers that report negative or null research outcomes. From this perspective, the paper can be accepted after a major revision that highlights the negative aspect of the outcome.

My strongest objection pertains to the concept of using autoregressive machine learning (AR ML) models to predict the epidemic of an emerging disease caused by an entirely new pathogen. While AR ML models may perform well in predicting smooth, predictable periods of an epidemic, as demonstrated by the authors, they necessarily fail at critical moments. Specifically, these models do not adequately address the system's response to planned restrictions, the height of epidemic waves: registered and more importantly the real ones, the dark figure (detection) coefficients, and the real derivatives of the former e.g. the real required number of hospital beds during peak demand periods.

Selecting performance metrics based on the average performance across numerous segments, predominantly "smooth periods" with only occasional "tough periods," creates a misleading impression of the model's overall efficacy. For established diseases, this approach might be acceptable since high waves with high mortality are not typically expected. In such cases, precise predictions of stationary and regular time series, as seen in seasonal disease profiles (e.g., influenza and SARI in non-pandemic years), provide useful information to stakeholders.

However, for non-stationary, emerging disease epidemics (such as the COVID-19 pandemic), AR ML models cannot be equated with other, more mechanistic and explainable models. Properly constructed mechanistic models (and some AI models based on a broader set of information) are developed on purpose to predict and explain non-stationary, reactive, complex, and socially significant periods in an epidemic. Contrary to the authors' statements in lines 49 and 50, these models are designed for such purposes.

Using AR ML models to predict an emerging disease epidemic is akin to driving a car without fastening the seat belts. Most of the time, this approach may seem sufficient, but it fails at the accident. Mechanistic models, and other models that incorporate factors beyond autoregressive parameters, in this metaphor, function like seat belts: while they may be uncomfortable or costly to maintain during smooth periods, they have the potential to be effective during crises (e.g., the emergence of a new variant, reaching herd immunity, or the implementation of restrictions).

### Detailed Explanation:

I realize that my main remark might be unsatisfactory for the authors; therefore, I provide a more detailed explanation of my point of view:

The underlying epidemic processes for these two groups of diseases (namely COVID-19 during the pandemic years and Influenza and SARI) are fundamentally different. COVID-19 was an emerging disease caused by a completely new pathogen, with no pre-existing immunity in any human population. This made the COVID-19 epidemic process entirely non-stationary from the onset until herd immunity was achieved. This is a switch like process, which was to some extend controlled by various mitigation policies.. Additionally, the manifestation of this process in the observed number of cases, hospitalizations, and deaths was irregular, as these quantities were influenced not only by the epidemic process itself but also by random and unpredictable administrative changes in survey systems implemented independently by state and regional authorities.

As I argued initially, using autoregressive ML models (broadly understood, including AR, MVAR, ML, and similar models) for predicting the course of an emerging disease is misleading unless it is explicitly stated that such models, by their construction, cannot forecast critical events in the pandemic's trajectory. Indeed, the accuracy of these models, averaged over all time segments, might be high and seemingly satisfactory. They perform well during relatively long periods of "smooth times" in a pandemic when no interventions are instituted, no new variants are present, and no changes in survey methods occur. However, during the most crucial moments, when decision-makers and resource managers need accurate predictions, autoregressive ML models inevitably fail. These models lack information about the system's potential response to new factors. In contrast, mechanistic models aim to predict system behaviour during non-stationary periods driven by new, previously unknown factors, such as the initiation of a vaccination campaign or the implementation of a strict lockdown.

It is inaccurate, as the authors have stated, that mechanistic models are dependent on historical trend data (lines 49-50). Mechanistic models, in general, do not rely on historical trend data provided there is sufficient knowledge about the fundamental parameters they use. Importantly, mechanistic models do not require the assumption of process stationarity. The accuracy of well-constructed mechanistic models (such as Agent-Based Models) might be comparable or slightly weaker during "smooth times," but they are designed specifically to predict system reactions to factors mentioned by the authors (lines 50-52) without needing the assumption of stationarity.

In summary, autoregressive models (including ML ensemble models) are suitable for predicting disease courses in their stationary phases (as demonstrated for SARI and influenza in non-pandemic years) or during "smooth times" within pandemic years. However, they are incapable of predicting crucial moments and factors (e.g., the height of a wave) for future pandemic developments during the emerging phase when biological or social interventions are applied. In theory, AI models might predict these moments but only by incorporating information from sources other than disease data time series, such as politicians' tendencies to institute a lockdown, which is extremely challenging to predict. While I do not discuss the quality and accuracy of different mechanistic models here, which can vary, I emphasize the essence of their construction and capabilities based on the information they incorporate. The degree of complexity of single AR ML model and the use of ensemble models do not change the situation.

The article focuses on short-term predictions, which might partially justify using AR ML models for emerging disease prediction. However, the fundamental limitations of such an approach must be discussed in more detail. In my opinion, the article is based on a misleading general idea that should be rephrased or thoroughly discussed.

### Minor Remarks on Specific Statements in the Article

**Line 57:** "In principle, ensemble models can be understood as a collection of rather simplistic base models."

*Comment:* Why describe the base models as simplistic? Each base model can be constructed very differently.

**Line 79:** "We incorporate six different datasets related to COVID-19."

*Comment:* What does "incorporate" mean here? The crucial question is whether the model depends on a single variable or is multivariate. This is not clearly explained in the article. Specifically, does the prediction of incident cases rely solely on past incident cases, or does it include other time-series data (e.g., the number of deaths)? Please provide detailed description of input layer to the model.

**Line 80:** "Additionally, we evaluated the models on weekly Influenza cases and weekly hospital admissions related to severe acute respiratory infections (SARI) in Germany."

*Comment:* Similar to the previous remark, what data does each model depend on? Are these models single-variable autoregressive or multivariable autoregressive?

*Line 82:* "For this, we followed a sliding window approach (see Figure 1) with a training window size of 70 days."

- **Comment:** The training window is 70 days long, but what is the size of the input data?

**Line 84:** "The objective was to forecast the value of the time series 14 days ahead of time."

*Comment:* The limitations of this objective need to be discussed. While this is suitable for short-term resource planning, it is not sufficient for long-term preparedness and intervention planning. More global measures are needed, such as wave height and the time of the wave peak.

**Line 86:** "The MAPE represents the deviation of the prediction from the real data."

*Comment:* These are not "real"

---

## [Decision Letter · Decision Letter 1]

25 Jul 2024

A dynamic ensemble model for short-term forecasting in pandemic situations

PGPH-D-24-00498R1

Dear Mr. Botz,

We are pleased to inform you that your manuscript 'A dynamic ensemble model for short-term forecasting in pandemic situations' has been provisionally accepted for publication in PLOS Global Public Health.

Best regards,

Francesco Branda, Ph.D.

Academic Editor

Reviewer Comments (if any, and for reference):

Reviewer's Responses to Questions

**Comments to the Author**

1. If the authors have adequately addressed your comments raised in a previous round of review and you feel that this manuscript is now acceptable for publication, you may indicate that here to bypass the “Comments to the Author” section, enter your conflict of interest statement in the “Confidential to Editor” section, and submit your "Accept" recommendation.

Reviewer #1: All comments have been addressed

Reviewer #2: All comments have been addressed

2. Does this manuscript meet PLOS Global Public Health’s publication criteria? Is the manuscript technically sound, and do the data support the conclusions? The manuscript must describe methodologically and ethically rigorous research with conclusions that are appropriately drawn based on the data presented.

Reviewer #1: Yes

Reviewer #2: Yes

3. Has the statistical analysis been performed appropriately and rigorously?

Reviewer #1: Yes

Reviewer #2: Yes

4. Have the authors made all data underlying the findings in their manuscript fully available (please refer to the Data Availability Statement at the start of the manuscript PDF file)?

Reviewer #1: Yes

Reviewer #2: Yes

5. Is the manuscript presented in an intelligible fashion and written in standard English?

Reviewer #1: Yes

Reviewer #2: Yes

6. Review Comments to the Author

Reviewer #1: My comments have been sufficiently taken into account.

Reviewer #2: Thank you for adding the clarifications and discussions to the revised manuscript. I have no further comments.

7. PLOS authors have the option to publish the peer review history of their article (what does this mean?). If published, this will include your full peer review and any attached files.

**Do you want your identity to be public for this peer review?** For information about this choice, including consent withdrawal, please see our Privacy Policy.

Reviewer #1: **Yes: **Franciszek Rakowski

Reviewer #2: No
